# Metabolic Imaging and Molecular Biology Reveal the Interplay between Lipid Metabolism and DHA-Induced Modulation of Redox Homeostasis in RPE Cells

**DOI:** 10.3390/antiox12020339

**Published:** 2023-01-31

**Authors:** Giada Bianchetti, Maria Elisabetta Clementi, Beatrice Sampaolese, Cassandra Serantoni, Alessio Abeltino, Marco De Spirito, Shlomo Sasson, Giuseppe Maulucci

**Affiliations:** 1Department of Neuroscience, Biophysics Sections, Università Cattolica del Sacro Cuore, Largo Francesco Vito, 1, 00168 Rome, Italy; 2Fondazione Policlinico Universitario “A. Gemelli” IRCCS, 00168 Rome, Italy; 3Institute of Chemical Sciences and Technologies “Giulio Natta” (SCITEC)—CNR, Largo Francesco Vito, 1, 00168 Rome, Italy; 4Faculty of Medicine, Institute for Drug Research, The Hebrew University, Jerusalem 911210, Israel

**Keywords:** diabetic retinopathy, lipid metabolism, β-oxidation, oxidative stress, docosahexaenoic acid (DHA), blood-retinal barrier, retinal diseases, human retinal pigment epithelium cells (ARPE-19), metabolic imaging

## Abstract

Diabetes-induced oxidative stress induces the development of vascular complications, which are significant causes of morbidity and mortality in diabetic patients. Among these, diabetic retinopathy (DR) is often caused by functional changes in the blood–retinal barrier (BRB) due to harmful oxidative stress events in lipids, proteins, and DNA. Docosahexaenoic acid (DHA) has a potential therapeutic effect against hyperglycemia-induced oxidative damage and apoptotic pathways in the main constituents of BRB, retinal pigment epithelium cells (ARPE-19). Effective antioxidant response elicited by DHA is driven by the activation of the Nrf2/Nqo1 signaling cascade, which leads to the formation of NADH, a reductive agent found in the cytoplasm. Nrf2 also induces the expression of genes encoding enzymes involved in lipid metabolism. This study, therefore, aims at investigating the modulation of lipid metabolism induced by high-glucose (HG) on ARPE-19 cells through the integration of metabolic imaging and molecular biology to provide a comprehensive functional and molecular characterization of the mechanisms activated in the disease, as well the therapeutic role of DHA. This study shows that HG augments RPE metabolic processes by enhancing lipid metabolism, from fatty acid uptake and turnover to lipid biosynthesis and β-oxidation. DHA exerts its beneficial effect by ameliorating lipid metabolism and reducing the increased ROS production under HG conditions. This investigation may provide novel insight for formulating novel treatments for DR by targeting lipid metabolism pathways.

## 1. Introduction

Diabetes mellitus (DM), a multifactorial systemic disease that affects millions of individuals worldwide, has become more common over recent years [1]. The main symptom and clinical indicator of DM are persistent high blood glucose levels (hyperglycemia) due to insulin deficiency and impaired peripheral glucose utilization in insulin-sensitive tissues [2]. DM is also associated with various metabolic abnormalities, including dyslipidemia [3], elevated levels of circulating blood free fatty acids, impaired lipid turnover [4], increased oxidative stress [5,6], and abnormal production of advanced glycation end products (AGEs) [7,8]. Together, these changes lead to the development of diabetes-related vascular complications, which disrupt the vasculature’s normal shape and physiology, causing gradual tissue and organ damage, dysfunction, and ultimately failure [9]. Nervous system damage (neuropathy), renal system damage (nephropathy), and eye damage (retinopathy) are caused by microvascular malfunction, [10]. Conversely, diabetes-associated macro vascular disease is manifested in peripheral vascular disease and cardiovascular disease. Studying the variables that trigger diabetes-induced vascular dysfunction may lead to the discovery of new strategies to reduce morbidity and mortality in subjects with diabetes-induced vascular dysfunction [9].

Several studies have shown that, among a wide range of chronic hyperglycemia-induced complications, such as inflammation [11,12], as well as the accumulation of intracellular reactive oxygen species (ROS), are leading causes of cellular damage. ROSs are very reactive radicals, ions, or molecules that are produced intracellularly by impaired mitochondrial functions or by interactions with exogenous sources. It is widely known that oxidative stress plays a significant role in the development of diabetic retinopathy (DR), which results in severe vision loss and blindness [13,14]. Indeed, being a part of the central nervous system (CNS), the retina is particularly vulnerable to changes in its microenvironment, whose maintenance is crucial to support the health and proper functionality of retinal cells [15]. Thus, impairments in the functionality of the blood–retinal barrier (BRB) has been directly related to the onset and progression of retinal diseases [16].

Retinal pigment epithelium (RPE) cells, situated between the neurosensory retina and the vascular choroids, have a crucial function in preserving the normal structural and functional integrity of the retina. These cells, which are characterized by a prevalently non-glycolytic metabolism [17], play an essential role in the retinal outer segment renewal by phagocytosis and digestion of shed outer segment tips. This phagocytic activity provides the RPE with lipids that may be either metabolized via β-oxidation to generate acetyl CoA or stored in the form of triglycerides (TAG) [18]. Being one of the most vulnerable cell populations of retina, and in view of the tight metabolic coupling with photoreceptors, RPE cell damage has been related to the development of DR, making them a useful and reliable model for researching the BRB’s functional changes brought on by diabetes [19].

Amongst several metabolic pathways that have been implicated in diabetes-induced vascular damage, the metabolic alteration of the lipid turnover has been proposed to play a key role in BRB disruption, but the mechanisms are not yet fully investigated [20]. A positive correlation between increased plasma lipid levels and the initiation and progression of DR has been observed in various clinical trials [21,22,23]. Of interest are the studies that show that the administration of ω3 polyunsaturated fatty acids (PUFAs) may reduce the risk of DR onset and further progression [24]. For instance, the anti-inflammatory and antioxidant properties of docosahexaenoic acid (DHA) on retinal cells have received considerable attention, making it a potential protective agent. 

In a previous work [25], we investigated the potential protective role of DHA at physiological concentrations [26] to prevent hyperglycemia-induced oxidative damage and apoptotic pathways in ARPE-19 cells. This study reveals that the effective antioxidant response elicited by DHA is driven by the activation of the Nrf2/Nqo1 signaling cascade, as well as the formation of the reductive agent NADH in the cytoplasm. Interestingly, Nrf2, which is known to be a master regulator of the intracellular antioxidant system as well as of cellular homeostasis, also reduces the expression of genes that encode enzymes involved in lipid metabolism, including β-oxidation and lipases [27]. From this perspective, to evaluate the effect on lipid turnover, we focused on different genes that are involved in several steps of lipid metabolic pathways, from fatty acids uptake (FATP1 and CD36) to lipid storage (DGAT1) and lipolysis (ATGL), including biosynthesis (SREBP) and β-oxidation (CPT1).

This study aimed at investigating alterations in lipid metabolism induced by high-glucose on ARPE-19 cells, as well as the potential ameliorating effects of DHA by combining metabolic imaging and gene expression of target proteins to provide a comprehensive functional and molecular characterization of mechanisms activated in the disease.

## 2. Materials and Methods

### 2.1. Establishment of the In-Vitro Model: Cells Culture and Treatments

Human retinal pigment epithelium (RPE) cells (ARPE-19) were purchased from the American Type Cell Culture (ATCC–CRL–2302, Manassas, VA, USA) and cultured in Advanced DMEM/F12 basal medium (Thermo Fisher Scientific, Inc., Waltham, MA, USA), supplemented with 20% fetal calf serum (FCS, Merck Life Science S.r.l., Milano, Italy) and 100 U/mL penicillin–streptomycin (Gibco™, Thermo Fisher Scientific, Inc., Waltham, MA, USA), with physiological glucose levels (5 mM). During growth, cells were maintained in a humidified environment (5% CO_2_). Further sub-cultures of cells (passage number 12) were performed at the proper density in accordance with each experimental method. To investigate the effect of the high glucose concentrations on ARPE-19 metabolism, cultured cells were treated as previously reported [25]. A 50 mM concentration of D-Glucose was added to cells for 20 h, followed by the administration of docosahexaenoic acid (DHA) for a further 16 h (Cayman Chemical, Ann Arbor, Michigan, MI, USA—50 mg in 200 µL ethanol), and this was then complexed to fatty acid-free bovine serum albumin (FAF-BSA, Merck Life Science S.r.l., Milano, Italy), prior to use as described [28], to reach the final concentration of 60 µM of DHA. 

### 2.2. Confocal Microscopy Imaging for the Quantification of Intracellular Non-Polar Aggregate

A Nikon A1-MP confocal microscope outfitted with a 2-photon Ti:Sapphire laser (Mai Tai, Spectra Physics, Newport Beach, CA, USA), emitting 80-fs pulses at a repetition rate of 80 MHz, was used to characterize the functional properties of ARPE-19 cells. An on-stage incubator (OKOLAB) kept a constant temperature of 37 °C and a 5% level of CO_2_. For the quantification of intracellular non-polar aggregates, the lipophilic probe Laurdan was used. Cells were treated with 1 µM of Laurdan. Laurdan intensity images (excitation: 740 nm) were recorded in the two emission ranges 450/50 nm and 525/50 nm with a 1024 × 1024-pixel resolution, and a 60× oil-immersion objective was used to visualize lipid aggregates. 

### 2.3. Isolation of RNA and RT-PCR for the Molecular Characterization of ARPE-19 Lipid Metabolism

Total RNA was extracted using the RNeasy MicroKit (Qiagen, Hilden, Germany), and its concentration was determined by spectrophotometric measurements at 280 and 260 nm. Using the Quan-tiTect Reverse Transcription Kit, the extracted total RNA was utilized to create the first strand of cDNA (Qiagen). The manufacturer’s instructions were followed while using PowerUpTM SYBR^®^ Green Master Mix (2X) reagents from Applied Biosystem in Waltham, MA, USA. The 7900HT FAST REAL-TIME PCR SYSTEM was used to quantify gene expression (Applied Biosystems, Waltham, MA, USA). Primers were purchased from Thermo Fisher Scientific, Inc. (Waltham, MA, USA). Each gene target quantification reaction was performed separately with the respective primer sets, as reported in Table 1.

According to the methodology reported in [25], a constant annealing temperature was employed for both the amplification and melt curve reaction settings.

Technical triplicates, no template controls (NTC), and samples for each run were used in triplicate separate runs for primer pair optimization and validation. The Applied Biosystem software (SDS 2.4.1, available online at https://www.thermofisher.com/it/en/home/technical-resources/software-downloads/applied-biosystems-7900ht-fast-real-timespcr-system.html, accessed on 29 January 2023) was used to evaluate the gene expression findings. Assuming the β-actin gene to be an endogenous and reference control, the average of the three threshold cycle values (*Ct*) were automatically filled in as follows:(1)$$\Delta C_{t}=C_{t}^{target}-C_{t}^{reference}$$
where $C_{t}^{target}$ is the target gene’s average of the three threshold cycle values, and $C_{t}^{reference}$ is the reference gene’s average threshold cycle value. Using the 2^−ΔΔCt^ technique, relative quantification was performed.

### 2.4. Statistical Analysis

Using Orange 3.32 (https://orangedatamining.com/, accessed on 29 January 2023) and Python 3.10 (https://www.python.org/, accessed on 29 January 2023) with the libraries *pandas* (https://pypi.org/project/pandas/, accessed on 29 January 2023), *numpy* (https://pypi.org/project/numpy/, accessed on 29 January 2023), *matplotlib* (https://pypi.org/project/matplotlib/, accessed on 29 January 2023), *seaborn* (https://pypi.org/project/seaborn/, accessed on 29 January 2023), and *scikit_posthocs* (https://pypi.org/project/scikit-posthocs/, accessed on 29 January 2023), Student’s *t*-tests were carried out on sets of biological and biophysical data. One-way ANOVA for parametric variables has been used to compare baseline features between samples. Then, for post-hoc comparisons among samples, Tukey’s test was employed.

## 3. Results

This study combined functional and molecular characterization of genes involved in lipid metabolism to unravel the impact induced by high-glucose levels on lipid metabolism and the ameliorating effect of DHA. To this aim, we selected different groups of genes involved in lipid metabolic pathways, whose expression can provide a clear insight into the mechanisms activated by the supplied treatments. A recent study reports the impact of high glucose concentrations on the growth rate of cultured ARPE-19 [25].

### 3.1. Effect of DHA as a Modulator of the High Glucose-Induced Enhancement of Fatty Acid Uptake

The first step of our analysis consists in the evaluation of two genes that regulate FA uptake, CD36 and FATP1. Mean values ± standard deviations are reported in Figure 1.

The fatty acid translocase, also known as cluster of differentiation 36 (CD36), as well as fatty acid transport 1 (FATP1), play a key role in lipid homeostasis since they regulate the coding of proteins that mediate cellular FFA uptake. In addition, CD36 is directly involved in the process of photoreceptors phagocytosis of the rod outer segments. Figure 1 depicts HG-induced over-expression of both gene products in comparison with CTRL cells (CD36 ~6-folds higher, Figure 1A, and FATP1, ~2-folds higher, Figure 1B). Interestingly, the addition of DHA to CTRL cells under normal glucose levels promoted the expression of CD36 to a similar extent as the effect of HG (6.17 ± 1.74 for HG and 5.12 ± 0.77 for DHA, respectively). Conversely, DHA inhibited FATP1 expression (0.07 *±* 0.01). When added together with high-glucose, DHA somewhat reduced the expression of CD36 from 6.17 ± 1.74 (HG) to 3.86 ± 0.35 (HG + DHA, mean ± sd, *p* < 0.05, (n = 3)). Conversely, DHA under HG conditions significantly inhibited FATP1 expression from 1.86 ± 0.72 (HG) to 0.60 ± 0.13 (HG + DHA).

### 3.2. Effect of DHA as a Modulator of the High Glucose-Induced Enhancement of Fatty Acid Turnover

The metabolism of intracellular fatty acids (FA) consists of a complex network of reactions that regulate lipid storage and usage. FAs in cells are stored in the form of TAG in specific and dynamic organelles, named lipid droplets (LD). The TAG can be further hydrolyzed and release FFA to supply energetic demand. To investigate the effect of high-glucose on the intracellular lipid turnover, we focused on the expression of diacylglycerol O-acyltransferase 1 (DGAT1) and adipose triglyceride lipase (ATGL), which are responsible for the regulation of balance between lipid storage and mobilization. DGAT1 encodes a protein, which catalyzes the conversion of diacylglycerol and fatty acyl CoA to TAG, while ATGL catalyzes the first reaction of lipolysis, where TAGs are hydrolyzed back to diacylglycerols. The results obtained are reported in Figure 2.

Figure 2A shows that the presence of high-glucose concentrations induces a marked over-expression of DGAT1 (17.19 ± 9.03) with respect to CTRL (*p* < 0.05), whereas the addition of DHA under the HG conditions restored it to the levels of expression of the CTRL state (11.10 ± 5.52 for HG + DHA). HG also augmented the expression of ATGL (Figure 2B) with respect to untreated cells (12.19 ± 2.20, *p* < 0.001). However, ATGL is reduced back to CTRL levels by the further treatment with DHA (0.80 ± 0.93, *p* < 0.001 with respect to HG alone).

### 3.3. Quantification of Intracellular Non-Polar Aggregates

We further exploited the lipophilic and solvatochromic properties of the fluorescent probe Laurdan [29] in combination with machine learning-based tools for pixel classification by confocal microscopic images [30] to quantify the intracellular distribution of non-polar (NP) aggregates [31,32,33]. Laurdan, due to its lipophilic properties, also localizes to intracellular lipid compartments, including LD [34,35,36]. Because Laurdan emission wavelength strictly depends on the polarity of its microenvironment, fluorescence from non-polar compartments peaks at 450 nm, and it is thus detected in the blue channel.

Representative Laurdan fluorescence emission images for CTRL, HG, DHA, and HG + DHA, respectively, are depicted in Figure 3A–D, along with the bar plot quantifying the fraction of NP aggregates €.

These images indicate that abundance of the non-polar aggregates (blue spots) increased in HG cells (Figure 3B) in comparison with the CTRL (Figure 3A). A higher enhancement is retrieved in presence of DHA alone (Figure 3C). In cells treated with DHA in HG conditions, non-polar aggregates are also enhanced at a similar extent (Figure 3D). To quantify the fraction of these aggregates, we evaluated the area occupied by blue spots with respect to the total cells surface and the results, represented as the mean values ± standard deviation, which are reported in the graph in Figure 3E. The bar plot in Figure 3E shows that high levels of glucose induced a significant growth in the fraction of non-polar compartments with respect to control ARPE cells, with values increasing from 0.04 ± 0.01 to 0.07 ± 0.01 for CTRL and HG, respectively (*p*-adj < 0.001). Interestingly, the addition of DHA in both physiologic and high-glucose conditions causes a four-fold increment of the non-polar fraction with respect to CTRL, with values of 0.14 ± 0.04 for DHA (*p*-adj < 0.001) and 0.13 ± 0.03 for HG + DHA, respectively. 

### 3.4. Effect of DHA as a Modulator of the High Glucose-Induced Enhancement of Fatty Acid Turnover

To further evaluate the effect of high-glucose and the modulation exerted by DHA on fatty acid biosynthesis, we quantified the expression of the sterol regulatory element-binding protein (SREBP). This gene is involved in the induction of lipid biosynthesis [37]. Mean values ± standard deviations are reported in the following graph (Figure 4).

Figure 4 shows a significant increase in the normalized expression of SREBP in presence of high-glucose concentrations (1.52 ± 0.14) with respect to CTRL, highlighting an enhancement in the cellular metabolism. Conversely, the addition of DHA, either in physiological or in HG conditions, downregulated the expression of SREBP with values of 0.72 ± 0.11, for DHA. Of interest is the observation that DHA abolished HG-induced augmented expression of SREBP and reduced it below the CTRL level, with a value of 0.70 ± 0.13 (HG + DHA). 

### 3.5. Effect of DHA as a Modulator of the High Glucose-Induced Enhancement of β-Oxidation

Carnitine palmitoyltransferase I (CPT1) is a protein associated with the outer mitochondrial membrane where it mediates the entry of long-chain fatty acids into the mitochondria, thus constituting an obligatory step in the process of β-oxidation. Normalized expression values of this gene are represented in the graph in Figure 5.

Figure 5 shows that the highest expression, approximately 40-fold higher than CTRL, is observed when DHA is added to the cells under physiological conditions. Interestingly, a similar effect on CPT1 expression is induced by HG, with a value of 26.14 ± 3.63. Yet, surprisingly, the treatment with DHA under HG conditions was significantly less effective in augmenting CPT1 expression, (*p* < 0.001).

## 4. Discussion

This study aimed to investigate the effect of high-glucose concentrations on lipid metabolism of RPE cells highlighting the interplay between metabolic changes induced by hyperglycemia-like conditions and DHA-induced modulation of redox homeostasis, as represented in the following Figure 6.

We found that high glucose levels caused a considerable stimulation of the whole lipid metabolism in RPE cells, ranging from FA absorption to lipolysis, as evidenced by the marked over-expression of various genes and receptors implicated in lipid metabolic pathways. The increased lipid metabolism triggered by HG conditions may be explained in terms of the role of glucose as a signaling biomolecule directly involved in oxidative and inflammatory pathways, which can potentially activate a cascade of molecular mechanisms [38,39]. We have previously shown that, under the same experimental conditions, ROS formation and apoptotic pathways are triggered by HG in ARPE cells [25]. Since one of the main functions of RPEs is to establish a bidirectional flux between choroid and photoreceptors, modulating water, nutrients, ion, and waste product exchange between the outer and inner part of the retina, any structural modification of RPE membrane can result in either accumulation of toxic substances or energy deprivation for photoreceptors due to the strong metabolic coupling. These impairments can lead to photoreceptors damage and can be directly involved in the onset and development of retinal degeneration, which ultimately results in vision loss following BRB breakdown, whose functional alterations have been observed to precede the onset of the pathology [15,40,41]. 

FA uptake processes are enhanced in HG cells, as shown by the increase in RNA levels of CD36 and FATP1 (Figure 1), which code for proteins that promote cellular FA absorption. DHA supplementation in HG conditions significantly reduces mRNA expression levels of both FATP1 and CD36 [42], thus modulating the uptake of FAs and toxic lipoproteins in the retina and contributing to the maintenance of BRB integrity and functionality. FA turnover is enhanced in HG conditions, as shown by the significant increase induced by HG in the expression of DGAT1 and ATGL (Figure 2), respectively, the main drivers of lipid storage and lipolysis pathway. Further addition of DHA in HG conditions decreased only ATGL expression. Metabolic imaging experiments confirmed these alterations, as we found that lipid storage processes increased in HG cells, but the treatment with DHA further enhanced this pathway, as shown by the area fraction of non-polar aggregates. The significant increase in non-polar aggregates observed in DHA-treated cells is not surprising. Indeed, when the cells are exposed to such levels of DHA, their only alternative to survive the cytotoxic and detergent effect of the free FA is to store it as TAG in lipid droplets [31]. Another process affected by HG is lipid biosynthesis, indirectly tested through the SREBP expression. SREBP increased in presence of high-glucose concentrations with respect to CTRL, indicating an enhancement of the lipid biosynthetic pathways. Again, the addition of DHA down-regulated the expression of SREBP contrasting the HG induced effect. Additionally, β-oxidation is affected by HG and modulated by DHA. β-oxidation, and the ROS generated as byproducts, are modulated by DHA, which regulated this process by reducing CPT1 expression.

In summary, HG increased RPE metabolic processes by enhancing fatty acid uptake and turnover, lipid biosynthesis, and also β-oxidation. We have previously shown that DHA also induced a potent antioxidant response in RPE cells under the same conditions, which activates the Nrf2/Nqo1 signaling cascade and causes the synthesis of the reductive coenzyme NADH in the cytoplasm. Therefore, DHA protects cells by both reducing oxidative species production, as shown in this work, and activating an antioxidant response, as shown in [25]. It is interesting to observe that, in view of their close correlation with the RPE, which is a crucial regulator of retinal function, the potential beneficial effects of other long-chain ω3-PUFA have also been investigated. In particular, in addition to DHA, eicosapentaenoic acid (EPA) has shown antioxidant properties and the capacity to protect RPE cells against the oxidative conditions associated with diabetic retinopathy, although by different mechanisms [43]. Interestingly, despite both having positive effects on ARPE-19 cells by lowering ROS generation and minimizing oxidative damage brought on by H_2_O_2_, this study highlighted that the effectiveness of PUFAs depends not only on whether they are administered alone or in combination with each other, but also on the formulation, with better results in triglyceride or phospholipid-based formulations compared with ethyl esters. In additions, it has already been shown that other compounds, such as punicalagin [44] or idebenone [45], can induce a protective and/or therapeutic response in RPE cells under oxidative stress conditions by activating the Nrf2 antioxidant pathway. 

An important limitation of this study, however, is the absence of lactate, a major substrate used by RPE cells to fuel fatty acid oxidation. Indeed, since these cells relies on the oxidative metabolism of the outer retina through the oxidation of fatty acids rather than glycolysis to support its metabolic needs, we speculate that in absence of lactate the level of oxidation would be lower than that retrieved in vivo. From this perspective, further research is needed to investigate the contribution of lactate in this in vitro model, as well as to highlight if Nrf2 could be a master regulator of the overall antioxidant response. Indeed, in several cell types, it was observed that, besides influencing the intracellular antioxidant system, Nrf2 is also responsible of inducing expression of genes, which encode enzymes involved in lipid metabolism, including β-oxidation and lipases [27]. Moreover, the principal actors of these pathways have to be analyzed also in terms of Western blots and cellular assays.

## 5. Conclusions

In conclusion, our study of the processes behind the BRB’s redox homeostasis impairment induced by high glucose levels and the recovery by the supplementation of DHA might offer a possible therapeutic target for an early intervention in its therapy. In this respect, metabolic therapies acting on lipid metabolism may enforce the action of antioxidants agents by amplifying the effects of existing or novel therapies.

## Figures and Tables

**Figure 1 antioxidants-12-00339-f001:**
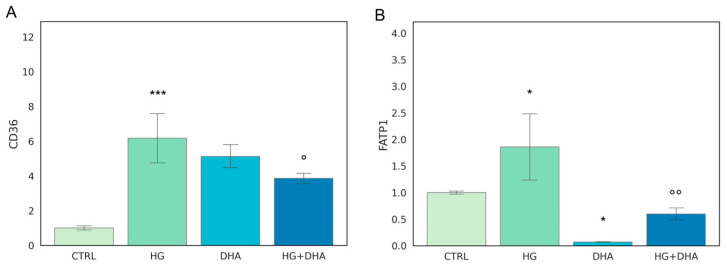
**Normalized expression of genes regulating fatty acid uptake.** Values of mRNA expression of CD36 (**A**) and FATP1 (**B**), normalized to expression levels of cells cultured in physiological conditions (CTRL = 1, light green), high glucose (50 mM, HG, for 36 h, dark green), DHA (60 μM, for 16 h, light blue), and HG + DHA (HG for 20 h and DHA for further 16 h, dark blue), respectively. Statistical results obtained from Tukey post-hoc comparisons among groups are shown along with the bar plot (* *p*-adj < 0.05 in comparison with CTRL; mean ± sd (n = 3); *** *p*-adj < 0.001 in comparison with CTRL; ° *p*-adj < 0.05 in comparison with HG alone; °° *p*-adj < 0.01 in comparison with HG alone).

**Figure 2 antioxidants-12-00339-f002:**
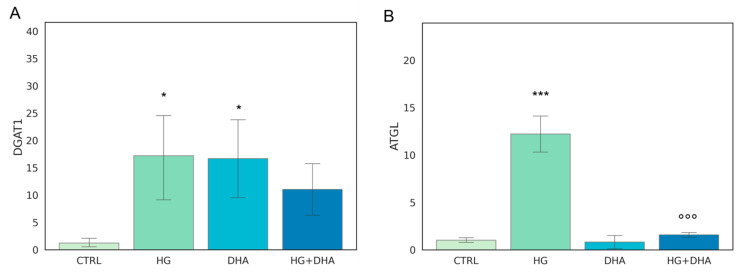
**Normalized expression of genes regulating FA turnover.** Values of mRNA expression of DGAT1 (**A**) and ATGL (**B**), normalized to expression levels of cells cultured in physiological conditions (CTRL = 1, light green), high glucose (50 mM, HG, for 36 h, dark green), DHA (60 μM, for 16 h, light blue), and HG + DHA (HG for 20 h and DHA for further 16 h, dark blue), respectively. Statistical results obtained from Tukey post-hoc comparisons among groups are shown along with the bar plot (* *p*-adj < 0.05 in comparison with CTRL; mean ± sd (n = 3); *** *p*-adj < 0.001 in comparison with CTRL; °°° *p*-adj < 0.001 in comparison with HG alone).

**Figure 3 antioxidants-12-00339-f003:**
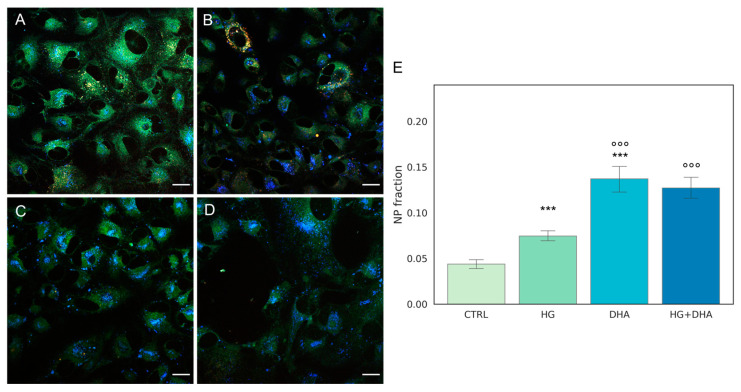
**Quantification of intracellular non-polar aggregates.** Laurdan fluorescence emission images for cells cultured in physiological conditions (CTRL, (**A**)), cells treated with glucose 50 mM (HG, for 36 h, (**B**)), cells treated with 60 µM (DHA, for 16 h, (**C**)), and cells with 50 mM glucose and further addition of 60 µM DHA (HG + DHA, (**D**)), respectively. Fluorescence emission was collected in three separate channels (em: 450/50 nm, 525/50 nm, and 595/50 nm). Non-polar aggregates are characterized by lower wavelength emission, resulting in blue spots in the images. Scale bar is 20 µm. The bar plot €(**E**) represents the fraction of intracellular non-polar aggregates for CTRL (light green), HG (36 h, dark green), DHA (16 h, light blue), and HG + DHA (HG for 20 h and DHA for further 16 h, dark blue), respectively. On the *y*-axis, the NP fraction is shown as mean ± sd. Along with the bar plot, the statistical Tukey post-hoc comparison across groups is displayed (*** *p*-adj < 0.001 with respect to CTRL, °°° *p*-adj < 0.001 in comparison with HG alone).

**Figure 4 antioxidants-12-00339-f004:**
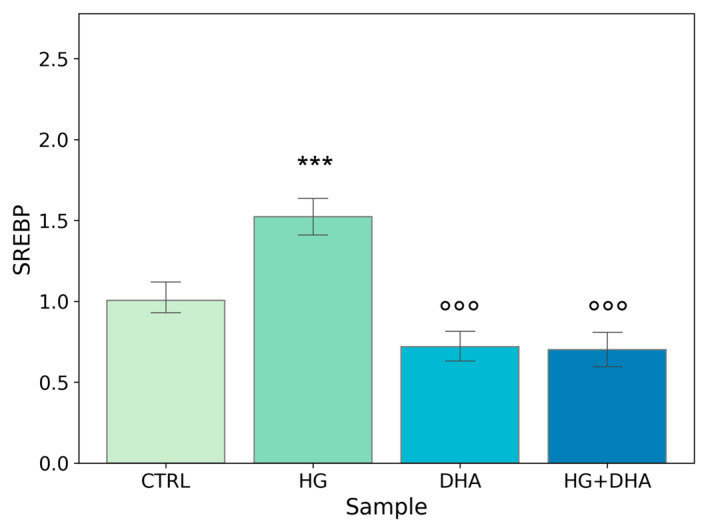
**Normalized expression of gene regulating FA biosynthesis.** Values of mRNA expression of SREBP, normalized to expression levels of cells cultured in physiological conditions (CTRL = 1, light green), high glucose (50 mM, HG, for 36 h, dark green), DHA (60 μM, for 16 h, light blue), and HG + DHA (HG for 20 h and DHA for further 16 h, dark blue), respectively. Along with the bar plot, statistical findings from the Tukey post-hoc comparison across groups are displayed (*** *p*-adj < 0.001 in comparison with CTRL; mean ± sd (n = 3); °°° *p*-adj < 0.001 in comparison with HG alone).

**Figure 5 antioxidants-12-00339-f005:**
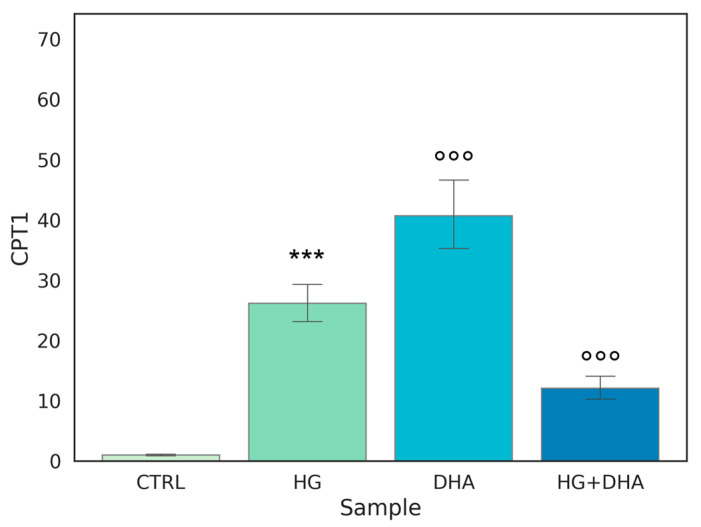
**Normalized expression of gene regulating β-oxidation.** Values of mRNA expression of CPT1, normalized to expression levels of cells cultured in physiological conditions (CTRL = 1, light green), high glucose (50 mM, HG, for 36 h, dark green), DHA (60 μM, for 16 h, light blue), and HG + DHA (HG for 20 h and DHA for further 16 h, dark blue), respectively. Along with the bar plot, statistical findings from the Tukey post-hoc comparison across groups are displayed (*** *p*-adj < 0.001 in comparison with CTRL; mean ± sd (n = 3); °°° *p*-adj < 0.001 in comparison with HG alone).

**Figure 6 antioxidants-12-00339-f006:**
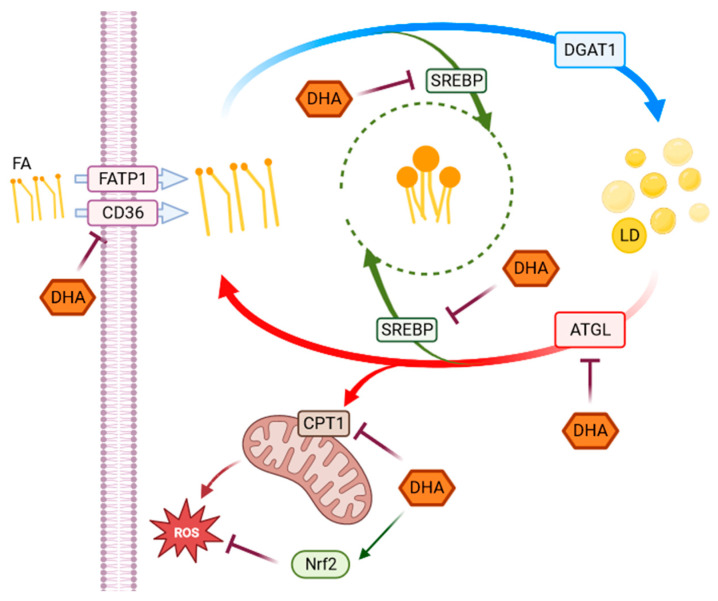
**Representative scheme of pathways activated in response to HG.** The scheme represents the observed metabolic and antioxidant pathways activated in response to HG highlighting the effects of DHA. Fatty acids (FA), whose uptake is modulated by CD36 and FATP1 through a series of chemical reactions, are stored in the form of TAG in lipid droplets. When required, TAG accumulated in LD can be broken down to be used in energy production. Several genes and enzymes regulate the lipid turnover, modifying the destination of intracellular lipids depending on cellular requirements. In our experimental model, DHA shows a modulatory effect on changes in lipid metabolism, as well as in redox homeostasis induced by high-glucose concentrations in ARPE-19 cells. SREBP and Nrf2 (in green) are transcription factors indirectly acting on the highlighted functions or pathways.

**Table 1 antioxidants-12-00339-t001:** List of primers used for quantitative RT-PCR.

Gene Target	Accession Code	Primer Sequence Forward (5′ to 3′)	Primer Sequence Reverse (5′ to 3′)
Β-Actin	NM_001101.5	AAACTGGAACGGTGAAGGTG	GTGGCTTTTAGGATGGCAAG
CD36	NM_001001548.3	CTTTGGCTTAATGAGACTGGGAC	GCAACAAACATCACCACACCA
FATP1	NM_198580.3	CTGCCCTTAAATGAGGCAGTCT	AACAGCTTCAGAGGGCGAAG
DGAT1	NM_012079.6	CGGGTCCGAGGGTGTCAATA	TCCACACAGCTCTGGCACTC
ATGL	JF279441.1	GCTTCCTCGGCGTCTACTAC	CAATGAACTTGGCACCAGCC
SREBP	NM_001005291.3	CTGGTCTACCATAAGCTGCAC	GACTGGTCTTCACTCTCAATG
CPT1	BT009791.1	ATCAATCGGACTCTGGAAACGG	TCAGGGAGTAGCGCATGGT

## Data Availability

The data are contained within the article.

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
