# Peer review of "Metabolic Imaging and Molecular Biology Reveal the Interplay between Lipid Metabolism and DHA-Induced Modulation of Redox Homeostasis in RPE Cells"

_antioxidants, 2023, doi:10.3390/antiox12020339_

Round 1

Reviewer 1 Report

In this manuscript, Bianchetti et al. and colleagues investigate the therapeutic potential of DHA in hyperglycemia-induced blood-retinal-barrier using human retinal pigment epithelial cells (ARPE-19) as an in vitro model. The authors utilize confocal imaging and PCR techniques to investigate the effect of DHA on ARPE-19 cells. Finally, the authors conclude that DHA alleviates the effect of hyperglycemia-induced oxidative damage by modulating lipid metabolism in ARPE-19 cells.

Major points

1.    In this reviewer's opinion, the main limitation of this study is the lack of in-depth molecular analyses to elucidate the key regulators that DHA modulates.

2.    It is unclear why the authors chose such an unphysiological concentration of high glucose to induce oxidative stress? Could the authors provide a detailed explanation?

3.    In Figure 3C-D, why is there a significant increase in non-polar aggregates, particularly DHA-treated cells? The authors should provide a detailed discussion about this in the discussion section.  

4.    In line with this, the authors should provide information on the effect of DHA on cell viability.

5.    It is well established that an increase or decrease in gene expression doesn’t always correlate with protein translation. The authors should provide supporting information on protein expression.

Minor point.

Page 1, line 21, check for the typo error.

Reviewer 2 Report

The retinal pigment epithelium (RPE), a monolayer of post-mitotic polarized epithelial cells strategically situated between the photoreceptors and the choroid, is the primary caretaker of photoreceptor health and function. The RPE is the gatekeeper of the outer retina and regulates the flux of nutrients and oxygen between the choroid and the subretinal space.

The RPE oxidizes lactate and fatty acids sparing glucose for the outer retina. The glucose transporter, GLUT1, in the basolateral and apical membranes of the RPE transports glucose from the choroid to the subretinal space. So, intracellular glucose levels are stable. Insulin is not needed for glucose transport. In the outer neural retina, glucose is metabolized through aerobic glycolysis producing large amounts of lactate.

In addition to lactate, the RPE utilizes fatty acids from ingested outer segments to fuel fatty acid oxidation and ketogenesis, generating β-hydroxybutyrate (β-HB) that is transported to the subretinal space. β-HB is taken up by photoreceptors and used to support oxidative metabolism. The metabolism of the RPE has been adapted to accommodate the ingestion of large amounts of lipids and proteins.

Moreover, from the literature, it is known that in vivo, this arrangement can impact its metabolism and has not yet been recapitulated in vitro. Culture media are usually high in glucose but not lactate or fatty acids, which likely affects the metabolome, transcriptome, proteome, and epigenome of cultured RPE.

So, it needs to be clarified in the current experiment how the high glucose and FA levels in the outer space of PRE could induce increased lipid metabolism in the absence of lactate. The RPE relies on the oxidative metabolism of the outer retina through the oxidation of fatty acids rather than glycolysis to support its metabolic needs. But, in the absence of lactate could be concluded that oxidation would be less than oxidation in vivo. The authors should add that limitation to the manuscript.

Docosahexaenoic acid (DHA), mainly extracted from microalgae and fish oil, is one of the most important omega-3 polyunsaturated fatty acids and has significant health benefits. So,  Another point in the discussion session that needs to be added is the efficacy of other FA in cultured PRE compared to the current experiment.

The authors should clarify the differentiation stage of the used PRE.

In line 41, the sentence should be corrected by the authors. Diabetes Mellitus develops only in the case of insulin deficiency. Insulin resistance could be present, but diabetes mellitus could never occur.

So, data are intriguing for future studies, although the authors should add the limitations mentioned above.

Reviewer 3 Report

Review of the manuscript "Metabolic imaging and molecular biology reveal the interplay between lipid metabolism and DHA-induced modulation of redox homeostasis in RPE cells".

I would like to thank the authors for the excellent research and presentation of the results. The results of this study will motivate clinicians in the supplementation of DHA in patients with diabetes. The manuscript is similar to an article by the same authors published in the Journal of Antioxidants. The results are different, but the point is the same.  According to the instructions of the Journal Antioxidants, the reuse of text that is copied from another source must be between quotes and the source must be cited. If a study's design or the manuscript's structure or language has been inspired by previous works, these works must be explicitly cited. 

Round 2

Reviewer 2 Report

No other comments. The authors answered my questions